# Abstract Reasoning with Distracting Features

**Kecheng Zheng**
University of Science
and Technology of China
zkcys001@mail.ustc.edu.cn

**Zheng-jun Zha**[*]
University of Science
and Technology of China
zhazj@ustc.edu.cn

**Wei Wei**
Google Research
wewei@google.com

## Abstract

Abstraction reasoning is a long-standing challenge in artificial intelligence. Recent studies suggest that many of the deep architectures that have triumphed over other domains failed to work well in abstract reasoning. In this paper, we first illustrate that one of the main challenges in such a reasoning task is the presence of distracting features, which requires the learning algorithm to leverage counter-evidence and to reject any of the false hypotheses in order to learn the true patterns. We later show that carefully designed learning trajectory over different categories of training data can effectively boost learning performance by mitigating the impacts of distracting features. Inspired by this fact, we propose feature robust abstract reasoning (FRAR) model, which consists of a reinforcement learning based teacher network to determine the sequence of training and a student network for predictions. Experimental results demonstrated strong improvements over baseline algorithms and we are able to beat the state-of-the-art models by 18.7% in the RAVEN dataset and 13.3% in the PGM dataset.

## 1 Introduction

A critical feature of biological intelligence is its capacity for acquiring principles of abstract reasoning from a sequence of images. Developing machines with skills of abstract reasoning help us to improve the understandings of underlying elemental cognitive processes. It is one of the long-standing challenges of artificial intelligence research [3, 12, 31, 34]. Recently, Raven's Progressive Matrices (RPM), as a visual abstract reasoning IQ test for humans, is used to effectively estimate a model's capacity to extract and process abstract reasoning principles.

Various models have been developed to tackle the problem of abstract reasoning. Some traditional models [4, 24, 25, 26, 27, 29, 30] rely on the assumptions and heuristics rules about various measurements of image similarity to perform abstract reasoning. As Wang and Su [38] propose an automatic system to efficiently generate a large number using first-order logic. There has also been substantial progress in both reasoning and abstract representation learning using deep neural networks [14, 15, 34, 39]. However, these deep neural based methods simply adopt existing networks such as CNN [22], ResNet [11] and relational network [35] to perform abstract reasoning but largely ignore some of the reasoning's fundamental characteristics.

One aspect that makes abstract reasoning substantially difficult is the presence of distracting features in addition to the reasoning features that are necessary to solve the problem. Learning algorithms would have to leverage various counter-evidence to reject any false hypothesis before reaching the correct one. Some other methods [36, 37] design an unsupervised mapping from high-dimensional feature space to a few explanatory factors of variation that are subsequently used by reasoning models to complete the abstract reasoning task. Although these models boost the performance of abstract

---

[*]Corresponding author.
[1]Full code are available at `https://github.com/zkcys001/distracting_feature`.

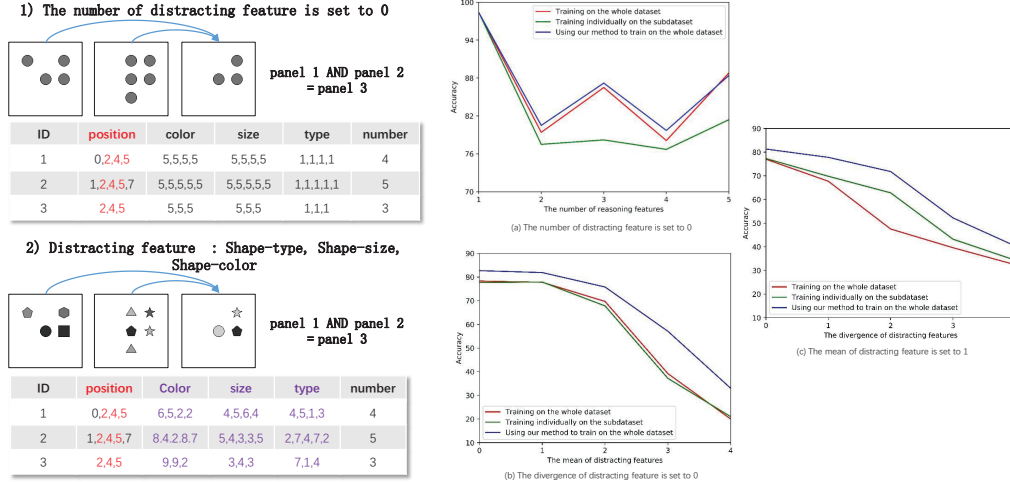

Figure 1: Left: Without distracting features, it is obvious to infer the abstract reasoning principles. Samples with distracting features confuse our judgment and make it harder to characterize reasoning features. Right: The influence of distracting features. (a) Without distracting features, training on the whole dataset is better than training on the individual dataset. (b) When the divergence of distracting features is set to zero, test performance decreases as the mean of distracting features increases. (c) When the mean of distracting features is set to one, test performance decreases as the divergence of distracting features increases.

Table 1: Test performance of LEN trained on different trajectories. "−>" denotes to training order. The first row demonstrates two datasets (i.e., 1 and 2) without distracting features while the second row illustrates datasets (i.e., 3 and 4) with distracting features. FRAR demonstrates our algorithm which optimizes learning trajectory to prevent distracting features from affecting the training of learning algorithms.

| Dataset | 1 | 2−>1 | 2 | 1−>2 | 1+2 | 1−>2 −>1+2 | 1−>2−> 1−>1+2 | 2−>1−> 2−>1+2 | FRAR |
|---|---|---|---|---|---|---|---|---|---|
| Acc(%) | 74.2 | 79.5 | 60.2 | 77.5 | 81.0 | 81.8 | 81.5 | 81.3 | **82.1** |

| dataset | 3 | 4−>3 | 4 | 3−>4 | 3+4 | 3−>4 −>3+4 | 3−>4−> 1−>3+4 | 4−>3−> 4−>3+4 | FRAR |
|---|---|---|---|---|---|---|---|---|---|
| Acc(%) | 52.5 | 58.4 | 64.5 | 65.9 | 58.2 | 61.0 | 59.6 | 62.1 | **67.6** |

reasoning tasks by capturing the independent factors of variation given an image, it is still difficult to find the reasoning logic from independent factors of variation and separate distracting features and reasoning features. Figure 1 shows one such example of abstract reasoning with distracting features where the true reasoning features in 1) is mingled with distracting ones in 2). Distracting features disrupt the learning of statistical models and make them harder to characterize the true reasoning patterns. On the right panel of Figure 1, we see that when we add more distracting features into the dataset (either through increasing the mean number of distracting features or through increasing the divergence of such features), the learning performance decrease sharply alert no information that covers the true reasoning patterns have been changed. Another observation with the distracting feature is that when we divide the abstract reasoning dataset into several subsets, training the model on the entire dataset would benefit the model as opposed to training them separately on the individual dataset. This is not surprising since features that are not directly benefiting its own reasoning logic might benefit those from other subsets. When distracting features are present, however, we see that some of the learning algorithms get worse performance when training on the entire dataset, suggesting that those distracting features trick the model and interfere with the performance.

To tackle the problem of abstract reasoning with distraction, we take inspirations from human learning in which knowledge is taught progressively according to a specific order as our reasoning abilities build up. Table 1 illustrates such an idea by dividing the abstract reasoning dataset into two parts as we change the proportion of datasets and take them progressively to the learning algorithm as learning proceeds. As we see from the results, when no distracting features are present (first row), changing the order of the training has little impacts on the actual results. When distracting features are present (second row), however, the trajectory of training data significantly affects the training outcome. The FRAR model that we propose to optimize training trajectory in order to prevent distracting features from affecting the training achieves a significant boost of 15.1% compares to training on a single dataset. This suggests that we are able to achieve better training performance by changing the order that the learning algorithm receives the training data.

The next question we want to ask is can we design an automated algorithm to choose an optimized learning path in order to minimize the adversarial impacts of distracting features in abstract reasoning. Some of the methods have been studied but with slightly different motivations. Self-paced learning [21] prioritize examples with small training loss which are likely not noising images; hard negative mining [28] assign a priority to examples with high training loss focusing on the minority class in order to solve the class imbalance problem. Mentornet[18] learns a data-driven curriculum that provides a sample weighting scheme for a student model to focus on the sample whose label is probably correct. These attempts are either based on task-specific heuristic rules, the strong assumption of a pre-known oracle model. However, in many scenarios, there are no heuristic rules, so it is difficult to find an appropriate predefined curriculum. Thus adjustable curriculum that takes into account of the feedback from the student accordingly has greater advantages. [10] leverages the feedback from the student model to optimize its own teaching strategies by means of reinforcement learning. But in [10], historical trajectory information is insufficiently considered and action is not flexible enough, lead to being not suitable for the situations where training trajectory should be taken into account.

In this paper, we propose a method to learn the adaptive logic path from data by a model named feature robust abstract reasoning model (FRAR). Our model consists of two intelligent agents interacting with each other. Specifically, a novel Logic Embedding Network (LEN) as the student model is proposed to disentangle abstract reasoning by explicitly enumerating a much larger space of logic reasoning. A teacher model is proposed to determine the appropriate proportion of teaching materials from the learning behavior of a student model as the adaptive logic path. With the guidance of this adaptive logic path, the Logic Embedding Network enables to characterize reasoning features and distracting features and then infer abstract reasoning rules from the reasoning features. The teacher model optimizes its teaching strategies based on the feedback from the student model by means of reinforcement learning so as to achieve teacher-student co-evolution. Extensive experiments on PGM and RAVEN datasets have demonstrated that the proposed FRAR outperforms the state-of-the-art methods.

## 2 Related Work

**Abstract reasoning**   In order to develop machines with the capabilities to underlying reasoning process, computational models [4, 24, 25, 26, 27, 29, 30] are proposed to disentangle abstract reasoning. Some simplified assumptions[4, 25, 26, 27] are made in the experiments that machines are able to extract a symbolic representation of images and then infer the corresponding rules. Various measurements of image similarity [24, 29, 30] are adopted to learn the relational structures of abstract reasoning. These methods rely on assumptions about typical abstract reasoning principles. As Wang and Su [38] propose an automatic system to efficiently generate a large number of abstract reasoning problems using first-order logic, there are substantial progress in both reasoning and abstract representation learning in neural networks. A novel variant of Relation Network [35] with a scoring structure [34] is designed to learn relational comparisons between a sequence of images and then reasoning the corresponding rules. Hill et al. [14] induce analogical reasoning in neural networks by contrasting abstract relational structures. Zhang et al. [39] propose a dynamic residual tree (DRT) that jointly operates on the space of image understanding and structure reasoning.

**Curriculum learning**   The teaching strategies of weighting each training example have been well studied in the literature[5, 6, 18, 21, 28, 32]. Self-paced learning [9, 16, 17, 21] prioritizes examples with small training loss which are likely not noising images; hard negative mining [28] assigns a

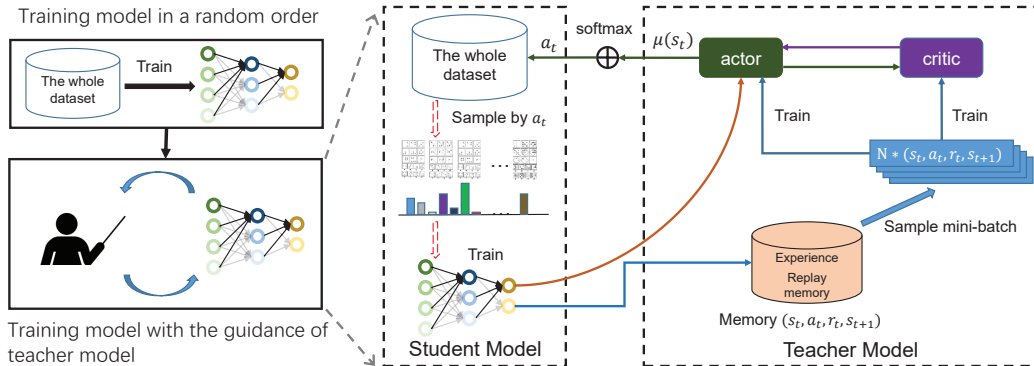

Figure 2: Overview of the interactive process between teacher model and student model. Left: The guidance of a teacher model replaces that training student model in random order. Right: Form the teacher model as a reinforcement learning problem. Our reinforcement learning agent (DDPG) receives the state $s_t$ from the performance of the student model and outputs a proportion of $a_t$ of training data at $t^{th}$ time step. After training the student model, the accuracy of the student model on a held-out validation set is evaluated as a reward $r$ which is returned to the reinforcement learning agent.

priority to examples with high training loss focusing on the minority class in order to solve the class imbalance problem. MentorNet [18] learns a data-driven curriculum that provides a sample weighting scheme for StudentNet focusing on the samples whose label are probably correct. These attempts are either based on task-specific heuristic rules or the strong assumptions of a pre-known oracle model. Fan et al. [10] leverage the feedback from a student model to optimize its own teaching strategies by means of reinforcement learning, so as to achieve teacher-student co-evolution. The re-weighting method [32] determines the example weights by minimizing the loss on a clean unbiased validation set.

**Disentangled Feature Representations** Disentangled feature representations efficiently encode high-dimensional features about the sensitive variation in single generative factors, isolating the variation about each sensitive factor in a fewer dimension. The key idea about disentangled representations is that real-world data mostly are generated by a few explanatory factors of variation which can be recovered by unsupervised learning algorithms. Hence, disentangled representations that capture these explanatory factors are expected to help in generalizing systematically [8, 19]. The sampling method based on disentangled representations is more efficient [13] and less sensitive to nuisance variables [33]. In terms of systematic generalization [1, 7], VASE [1] detects the adaptive shift of data distribution based on the principle of minimum description length, and allocates redundant disentangled representations to new knowledge. In other cases, however, it is not clear whether the gains of experiments are actually due to disentanglement [20]. In the abstracting reasoning tasks, some works [36, 37] learn an unsupervised mapping from high-dimensional feature space to a lower dimensional and more structured latent space that is subsequently used by reasoning models to complete reasoning task.

## 3 Feature Robust Abstract Reasoning

Our feature robust abstract reasoning algorithm is employed based on a student-teacher architecture illustrated in 2. In this architecture, the teacher model adjusts the proportions of training datasets and sends them to the student model. After these data are consumed, a student model will return its validation accuracy on the current batch which is used as rewards for the teacher model to update itself and to take the next action. This process repeats until the two models are converged.

### 3.1 Teacher Model

Since the rewards are generated by a non-differential function of the actions, we will use reinforcement learning to optimize the teacher model in a blackbox fashion.

**Action** We assume that each training sample is associated with a class label. In our dataset, this is taken to be the category of the abstraction reasoning. Those categories are a logic combination of some of the basic types such as "shape", "type" or "position". One such example can be seen in Figure 1 where "position and" is labeled as the category of the problem. Here we divide the training data into $C$ parts: $\mathcal{D} = (\mathcal{D}_1, \mathcal{D}_2, ..., \mathcal{D}_C)$, with each of the subset $\mathcal{D}_c$ denotes a part of the training data that belongs to category $c$. Here $C$ is the number of categories in the dataset. The action $a_t = < a_{t,1}, a_{t,2}, ..., a_{t,C} >$ is then defined to be a vector of probabilities they will present in the training batch. Samples in the training batch $x_i$ will be drawn from the dataset $\mathcal{D}$ from distribution $a_t$. $B$ independent draws of $x_i$ will form the mini-batch $< x_1, x_2, ...x_B >$ that will be sent to the student for training.

**State** The state of teacher model tracks the progress of student learning through a collection of features. Those features include:

1. Long-term features: a) the loss of each class over the last N time steps; b) validation accuracy of each class over N time steps;

2. Near-term features: a) the mean predicted probabilities of each class; b) the loss of each class; c) validation accuracy of each class; d) the average historical training loss; e) batch number and its label category of each class; f) action at the last time step; g) the time step.

**Reward** Reward $r_t$ measures the quality of the current action $a_t$. This is measured using a held-out validation set on the student model.

**Implementation** We use the deep deterministic policy gradient (DDPG) for continuous control of proportions of questions $a_t$. As illustrated in Figure 2, the teacher agent receives a state $s_t$ of student model at each time step $t$ and then outputs a proportion of questions as action $a_t$. Then, the student model adopts the proportion $a_t$ to generate the training data of $t^{th}$ time step. We use policy gradient to update our DDPG model used in the teacher network.

## 3.2 Logic Embedding Network

We can choose any traditional machine learning algorithms as our student model. Here, we propose a novel Logic Embedding Network (LEN) with the reasoning relational module which is more fitted for abstract reasoning questions, since it enables to explicitly enumerate a much larger space of logic reasoning. In the case of $N \times N$ matrices of abstract reasoning tasks, the input of LEN consists of $N^2 - 1$ context panels and $K$ multiple-choice panels, and we need to select which choice panel is the perfect match for these context panels. In the LEN, the input images firstly are processed by a shallow CNN and an MLP is adopted to achieve $N^2 - 1$ context embeddings and $K$ multiple-choice embeddings. Then, we adopt the reasoning module to output the score of combinations of given choice embeddings and $N^2 - 1$ context embeddings. The output of reasoning module is a score $s_k$ for a given candidate multiple-choice panel, with label $k \in [1, K]$:

$$s_k = f_\Phi \Big( \sum_{(x_{i_1}, x_{i_2}, ..., x_{i_N}) \in \chi_{k_1}} g_{\theta_1}(x_{i_1}, x_{i_2}, ..., x_{i_N}, z) + \sum_{(x_{j_1}, x_{j_2}, ..., x_{j_N}) \in \chi_{k_2}} g_{\theta_2}(x_{j_1}, x_{j_2}, ..., x_{j_N}, z) \Big),$$

(1)

where $\chi_k$ is the whole combinations of panels, $\chi_{k_1}$ is row-wise and column-wise combinations of panels and $\chi_{k_2} = \chi_k - \chi_{k_1}$ represents the other combinations of panels. $c_k$ is a embedding of $k^{th}$ choice panel, $x_i$ is a embedding of $i^{th}$ context panel, and $z$ is global representation of all 8 context embedding panels. For example, in the case of $3 \times 3$ matrices (N=3) of abstract reasoning tasks with 8 multiple-choice panels, $\chi_k = \{(x_i, x_j, x_k) | x_i, x_j, x_k \in S, S = \{x_1, x_2, ..., x_8, c_k\}, i \neq j, i \neq k, j \neq k\}$, $\chi_{k_1} = \{(x_1, x_2, x_3), (x_4, x_5, x_6), (x_7, x_8, c_k), (x_1, x_4, x_7), (x_2, x_5, x_8), (x_3, x_6, c_k)\}$ and $\chi_{k_2} = \chi_k - \chi_{k_1}$. $f_\Phi$, $g_{\theta_1}$ and $g_{\theta_2}$ are functions with parameters $\Phi$, $\theta_1$ and $\theta_2$, respectively. For our purposes, $f_\Phi$, $g_{\theta_1}$ and $g_{\theta_2}$ are MLP layers, and these parameters are learned by end-to-end differentiable. Finally, the option with the highest score is chosen as the answer based on a softmax function across all scores.

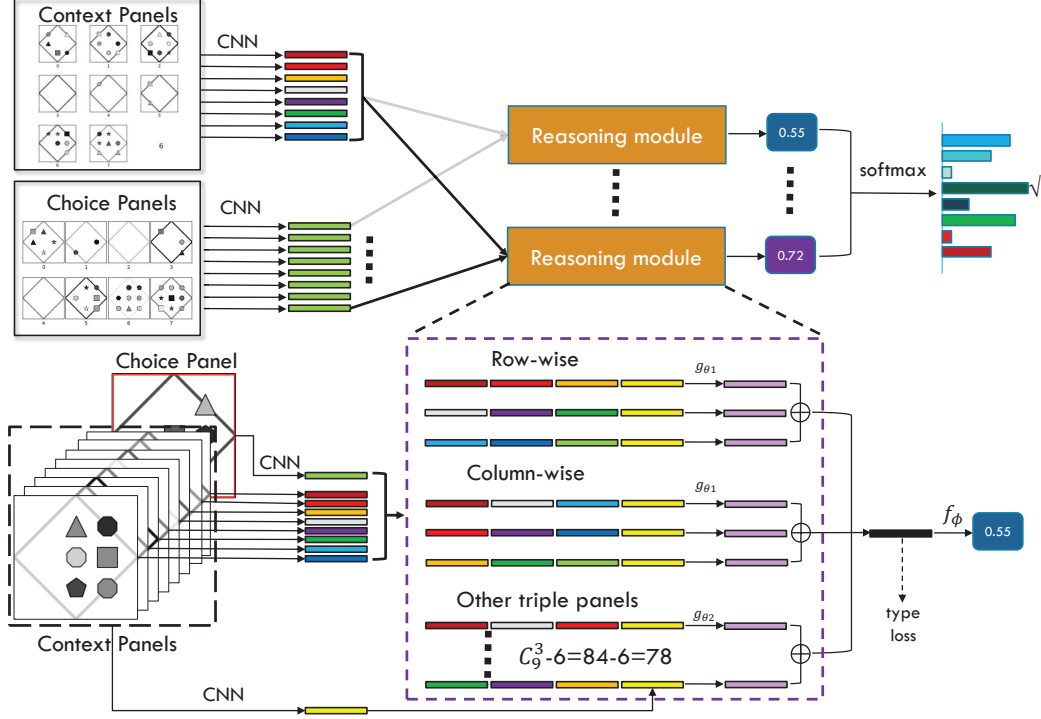

Figure 3: The architecture of Logic Embedding Network in the case of $3 \times 3$ abstract reasoning matrices with 8 multiple-choice panels. A CNN processes each context panel and each choice panel independently to produce 16 vector embeddings. Then we pass all 8 context embeddings with a choice embedding to a reasoning model, which enumerate the all space ($C_9^3 = 84$) of logic reasoning. And then this model outputs a score for the associated answer choice panel. There are totally 8 such reasoning module (here we only depict 1 for clarity) for each answer choice.

In abstract reasoning tasks, the goal is to infer reasoning logic rules that exist among N panels. Therefore, the structure of LEN model is very suitable for dealing with abstract reasoning task, since it adopts $g_{\theta_1}$ and $g_{\theta_2}$ to form representations of relationship of N panels, in the case of $3 \times 3$ matrices, including two context panels and a given multiple choice candidate, or triple context panels themselves. The function $g_{\theta_1}$ extracts the representations in row order and column order, such as "and" relational type in the color of shapes, while $g_{\theta_2}$ forms the representations of some reasoning logic rules regardless of order, such as the rule that all pictures contain common "shape". The function $f_{\Phi}$ integrates informations about context-context relations and context-choice relations together to provide a score of answer. For each multiple-choice candidate, our proposed LEN model calculates a score respectively, allowing the network to select the multiple-choice candidate with the highest score.

### 3.2.1 Two-stream Logic Embedding Network

During our training process, we have observed that "shape" and "line" features share little patterns in terms of logic reasoning. As a result, we have constructed a two-stream version of the logic embedding network in order to process these two types of features using its own parameters. Those two networks are then combined at the fusion layer before the predictions are generated.

## 4 Datasets

### 4.1 Procedurally Matrices dataset (PGM)

PGM [34] dataset consists of 8 different subdatasets, which each subdataset contains $119,552,000$ images and $1,222,000$ questions. We only compare all models on the neutral train/test split, which corresponds most closely to traditional supervised learning regimes. There are totally 2 objects

(Shape and Line), 5 rules (Progression, XOR, OR, AND, and Consistent union) and 5 attributes (Size, Type, Colour, Position, and Number), and we can achieve 50 rule-attribute combinations. However, excluding some conflicting and counterintuitive combinations (i.e., Progression on Position), we result in 29 combinations.

## 4.2   Relational and Analogical Visual rEasoNing dataset (RAVEN)

RAVEN [39] dataset consists of $1,120,000$ images and $70,000$ RPM questions, equally distributed in 7 distinct figure configurations: Center, $2 \times 2$ Grid, $3 \times 3$ Grid, Left-Right, Up-Down, Out-InCenter, and Out-InGrid. There are 1 object (Shape), 4 rules(Constant, Progression, Arithmetic, and Distribute Three) and 5 attributes(Type, Size, Color, Number, and Position), and we can achieve 20 rule-attribute combinations. However, excluding a conflicting combination (i.e., Arithmetic on Type), we result in 19 combinations.

# 5   Experiments

## 5.1   Performance on PGM Dataset

**Baseline Models**   We compare a comprehensive list of baseline models. From Table 2, we can see that CNN models fail almost completely at PGM reasoning tasks, those in include LSTM, CNN+MLP, ResNet-50, and W-ResNet.   The WReN model Barrett et al.  proposed [34] is also compared. Xander Steenbrugge et al.[36] explore the generalization characteristics of disentangled representations by leveraging a VAE modular on abstract reasoning tasks and can boost a little performance. Our proposed Logic Embedding Network (LEN) and its variant with two-stream (i.e.g, T-LEN) achieve a much better performance when comparing to baseline algorithms.

**Teacher Model Baselines**   We compare several baselines to our propose teacher model and adapt them using our LEN model.  Those baseline teacher model algorithms include curriculum learning, self-paced learning, learning to teach, hard example mining, focal loss, and Mentornet-PD. Results show that these methods are not effective in the abstract reasoning task.

Table 2: Test performance of all models trained on the neutral split of the PGM dataset. Teacher Model denotes that using the teacher model to determine the appropriate training trajectory. Type loss denotes that adding category label of questions into loss functions.

| Model | Acc(%) |
|---|---|
| LSTM[34] | 33.0 |
| CNN+MLP[34] | 35.8 |
| ResNet-50 [34] | 42.0 |
| W-ResNet-50 [34] | 48.0 |
| WReN [34] | 62.8 |
| VAE-WReN [36] | 64.2 |
| LEN | 68.1 |
| T-LEN | **70.3** |
| LEN + Curriculum learning[2] | 63.3 |
| LEN + Self-paced learning[21] | 57.2 |
| LEN + Learning to teach[10] | 64.3 |
| LEN + Hard example mining[28] | 60.7 |
| LEN + Focal loss[23] | 66.2 |
| LEN + Mentornet-PD [18] | 67.7 |
| WReN + type loss[34] | 75.6 |
| LEN + type loss | 82.3 |
| T-LEN + type loss | 84.1 |
| WReN + Teacher Model [34] | 68.9 |
| LEN + Teacher Model | 79.8 |
| T-LEN + Teacher Model | 85.1 |
| WReN + Teacher Model + type loss[34] | 77.8 |
| LEN + Teacher Model + type loss | **85.8** |
| T-LEN + Teacher Model +type loss | **88.9** |

**Use of Type Loss**   We have experimented by adding additional training labels into the loss function for training with WReN, LEN, and T-LEN. The improvements are consistent with what have been reported in Barrett's paper [34].

**Teacher Models**   Finally, we show that our LEN and T-LEN augmented with a teacher model achieve the testing accuracy above 79.8% and 85.1% respectively on the whole neutral split of the PGM Dataset. This strongly indicates that models lacking effective guidance of training trajectory may even be completely incapable of solving tasks that require very simple abstract reasoning rules. Training these models with an appropriate trajectory is sufficient to mitigate the impacts of distracting features and overcomes this hurdle. Further experiments by adding a type loss illustrate that teacher model and also be improved with the best performance of LEN (from 79.8% to 85.3%) and T-LEN (from 85.1% to 88.9%). Results with WReN with teacher network also reported improvements but is consistently below the ones with LEN and T-LEN models.

## 5.2   Performance on RAVEN Dataset

We compare all models on 7 distinct figure configurations of RAVEN dataset respectively, and table 3 shows the testing accuracy of each model trained on the dataset. In terms of model performance, popular models perform poorly (i.e., LSTM, WReN, CNN+MLP, and ResNet-50). These models lack the ability to disentangle abstract reasoning and can't distinguish distracting features and reasoning features. The best performance goes to our LEN containing the reasoning module, which is designed explicitly to explicitly enumerate a much larger space of logical reasoning about the triple rules in the question. Similar to the previous dataset, we have also implemented the type loss. However, contrary to the first dataset, type loss performs a bit worse in this case. This finding is consistent with what has been reported in [39]. We observe a consistent performance improvement of our LEN model after incorporating the teacher model, suggesting the effectiveness of appropriate training trajectory in this visual reasoning question. Other teaching strategies have little effect on the improvement of models. Table 3 shows that our LEN and LEN with teacher model achieve a state-of-the-art performance on the RAVEN dataset at 72.9% and 78.3%, exceeding the best model existing when the datasets are published by 13.3% and 18.7%.

Table 3: Test performance of each model trained on different figure configurations of the RAVEN dataset. Acc denotes the mean accuracy of each model, while other columns show model accuracy on different figure configurations. 2Grid denotes $2 \times 2$ Grid, 3Grid denotes $3 \times 3$ Grid, L-R denotes Left-Right, U-D denotes Up-Down, O-IC denotes Out-InCenter, and O-IG denotes Out-InGrid.

| model | Acc | Center | 2Grid | 3Grid | L-R | U-D | O-IC | O-IG |
|---|---|---|---|---|---|---|---|---|
| LSTM[39] | 13.1 | 13.2 | 14.1 | 13.7 | 12.8 | 12.5 | 12.5 | 12.9 |
| WReN[34] | 14.7 | 13.1 | 28.6 | 28.3 | 7.5 | 6.3 | 8.4 | 10.6 |
| CNN + MLP[39] | 37.0 | 33.6 | 30.3 | 33.5 | 39.4 | 41.3 | 43.2 | 37.5 |
| ResNet-18[39] | 53.4 | 52.8 | 41.9 | 44.2 | 58.8 | 60.2 | 63.2 | 53.1 |
| LEN + type loss | 59.4 | 71.1 | 45.9 | 40.1 | 63.9 | 62.7 | 67.3 | 65.2 |
| LEN | **72.9** | **80.2** | **57.5** | **62.1** | **73.5** | **81.2** | **84.4** | **71.5** |
| ResNet-18 + DRT [39] | 59.6 | 58.1 | 46.5 | 50.4 | 65.8 | 67.1 | 69.1 | 60.1 |
| LEN + Self-paced learning[21] | 65.0 | 70.0 | 50.0 | 55.2 | 64.5 | 73.9 | 77.8 | 63.8 |
| LEN + Learning to teach [10] | 71.8 | 78.1 | 56.5 | 60.3 | 73.4 | 78.8 | 82.9 | 72.3 |
| LEN + Hard example mining[28] | 72.4 | 77.8 | 56.2 | 62.9 | 75.6 | 77.5 | 84.2 | 72.7 |
| LEN + Focal loss[23] | 75.6 | 80.4 | 55.5 | 63.8 | 85.2 | 83.0 | 86.4 | 75.3 |
| LEN + Mentornet-PD[18] | 74.4 | 80.2 | 56.1 | 62.8 | 81.4 | 80.6 | 85.5 | 74.5 |
| LEN + Teacher Model | **78.3** | **82.3** | **58.5** | **64.3** | **87.0** | **85.5** | **88.9** | **81.9** |

## 5.3   Teaching Trajectory Analysis

We set two groups of experiments to examine training trajectory generated by the teacher model. In this setting, according to the rules of [34], we generate 4 subdatasets $(\mathcal{D}_1, \mathcal{D}_2, \mathcal{D}_3, \mathcal{D}_4)$, which will exhibit an "and" relation, instantiated on the attribute types of "shape". $\mathcal{D}_1$ denotes that we instantiate the "and" relation on the type of "shape" as reasoning attributes and does not set the distracting attribute. $\mathcal{D}_2$ denotes that the reasoning attribution is based on the "size shape" and do not set the distracting attribute. $\mathcal{D}_3$ is similar to $\mathcal{D}_1$, but "size" is set a random value as the distracting attribute. $\mathcal{D}_4$ is similar to $\mathcal{D}_2$, but "type" is set a random value as the distracting attribute. In summary, there not exist distracting attributes in $\mathcal{D}_1$ and $\mathcal{D}_2$. For $\mathcal{D}_3$ and $\mathcal{D}_4$, "size" and "type" are distracting

attributes respectively. We conduct experiments as follows. As shown table 1, in $\mathcal{D}_1$ and $\mathcal{D}_2$, the accuracy of joint training is higher than that of individual training. Without distracting attributes, $\mathcal{D}_1$ and $\mathcal{D}_2$ can promote each other to encode the reasoning attributes, thus improving the accuracy of the model. Adjusting the training trajectory in the dataset without distracting attributes only provides a small increase in the performance. It demonstrates that a model without the influence of distracting attributes is able to encode all the attributes into satisfactory embedding and perform abstract reasoning. However, joint training in the dataset $\mathcal{D}_3$ and $\mathcal{D}_4$ with distracting attributes do not promote each other. Experiments in table 1 show that training in an appropriate trajectory can effectively guide the model to encode a satisfactory attribution and improve the performance. Then, our proposed model is able to find a more proper training trajectory and achieve an obvious improvement.

## 5.4 Embedding Space Visualizations

To understand the model's capacity to distinguish distracting representations and reasoning representations, we analyzed neural activity in models trained with our logic embedding network. We generated 8 types of questions including 4 attributes: "position", "color", "type" and "size", as shown in Figure 4. Our model seems to encourage the model to distinguish distracting features and reasoning features more explicitly, which could in turn explain its capacity to disentangles abstract reasoning. We find that these activities clustered with the guidance of teacher model better than without it. It demonstrates that the adaptive path from teacher model can promote the model to characterize the reasoning features and distracting features, which is beneficial for abstract reasoning.

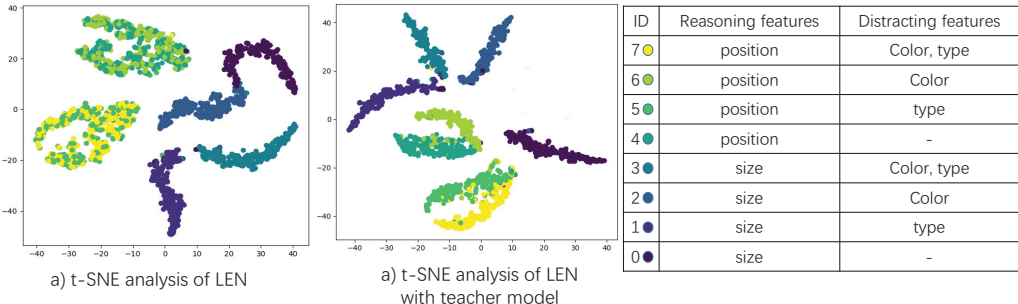

| ID | Reasoning features | Distracting features |
|----|--------------------|----------------------|
| 7 | position | Color, type |
| 6 | position | Color |
| 5 | position | type |
| 4 | position | - |
| 3 | size | Color, type |
| 2 | size | Color |
| 1 | size | type |
| 0 | size | - |

a) t-SNE analysis of LEN

a) t-SNE analysis of LEN with teacher model

Figure 4: t-SNE analysis of the last layer's embedding of logic embedding model. Each dot represents a (8-dimensional) state coloured according to the number of reasoning features and distracting features of the corresponding question.

## Conclusions

In this paper we proposed a student-teacher architecture to deal with distracting features in abstract reasoning through feature robust abstract reasoning (FRAR). FRAR performs abstract reasoning by characterizing reasoning features and distracting features with the guidance of adaptive logic path. A novel Logic Embedding Network (LEN) as a student model is also proposed to perform abstract reasoning by explicitly enumerating a much larger space of logic reasoning. Additionally, a teacher model is proposed to determine the appropriate proportion of teaching materials as adaptive logic path. The teacher model optimizes its teaching strategies based on the feedback from a student model by means of reinforcement learning. Extensive experiments on PGM and RAVEN datasets have demonstrated that the proposed FRAR outperforms the state-of-the-art methods.

## Acknowledgments

This work was supported by the National Key R&D Program of China under Grant 2017YFB1300201, the National Natural Science Foundation of China (NSFC) under Grants 61622211 and 61620106009 as well as the Fundamental Research Funds for the Central Universities under Grant WK2100100030.

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
