[Supplementary Material]

# Supplementary Material

**Kecheng Zheng**
University of Science
and Technology of China
zkcys001@mail.ustc.edu.cn

**Zheng-jun Zha** *
University of Science
and Technology of China
zhazj@ustc.edu.cn

**Wei Wei**
Google AI
wewei@google.com

## 1 Model Details

All the models are implemented in Pytorch with two NVIDIA GEFORCE GTX 1080 GPU. We set the learning rate to 0.01 and the batch size to 32.

### 1.1 Model Details of Logic Embedding Network

In this section, we provide details for our LEN we use to benchmark, including the exact hyper-parameter settings that we considered. Throughout this section, we will use the notation X-Y-Z to describe CNN and the notation w to describe MLP size. For a CNN, this notation refers to the hyper-parameters per layer: X refers to the kernel size, Y the stride, and Z the number of channels. For the MLP, it refers to the number of units per layer.

Table 1: CNN hyper-parameters of Logic Embedding Network

| local CNN | | global CNN | |
|---|---|---|---|
| Operator | Params | Operator | Params |
| Convolution | 3-2-32-3 | Convolution | 3-2-32-16 |
| BatchNorm | 3 | BatchNorm | 3 |
| ReLU | | ReLU | |
| Convolution | 3-2-32-3 | Convolution | 3-2-32-3 |
| BatchNorm | 3 | BatchNorm | 3 |
| ReLU | | ReLU | |
| Convolution | 3-2-32-3 | Convolution | 3-2-32-3 |
| BatchNorm | 3 | BatchNorm | 3 |
| ReLU | | ReLU | |
| Convolution | 3-2-32-3 | Convolution | 3-2-32-3 |
| BatchNorm | 3 | BatchNorm | 3 |
| ReLU | | ReLU | |
| Linear | 256 | Linear | 256 |

### 1.2 Two-stream Logic Embedding Network

The proposed Two-stream Logic Embedding Network consists of three major steps: firstly, we take advantage of an encoder-decoder model widely used in semantic segmentation tasks to learn the disentangled representations. The encoder-decoder model is a standard U-Net architecture based ResNet-50 with skipped connections, adopting the conditional random field as post-processing so as to achieve completer objects. This de-rendering model is only a pre-processing step to separate shapes and lines in the input image. After obtaining shape images and line images, we use the two

---

Table 2: MLP hyper-parameters of Logic Embedding Network

| $g_{\theta_1}$ | | $g_{\theta_2}$ | | $f_\phi$ | |
|---|---|---|---|---|---|
| Operator | Params | Operator | Params | Operator | Params |
| Linear | 1024 | Linear | 1024 | Linear | 512 |
| BatchNorm | 1024 | BatchNorm | 1024 | BatchNorm | 512 |
| ReLU | | ReLU | | ReLU | |
| Linear | 768 | Linear | 768 | Linear | 256 |
| BatchNorm | 768 | BatchNorm | 768 | BatchNorm | 256 |
| ReLU | | ReLU | | ReLU | |
| Linear | 512 | Linear | 512 | Dropout | 0.5 |
| BatchNorm | 512 | BatchNorm | 512 | Linear | 1 |
| ReLU | | ReLU | | | |
| Linear | 512 | Linear | 512 | | |
| BatchNorm | 512 | BatchNorm | 512 | | |
| ReLU | | ReLU | | | |

kinds of images to train two LEN models as two stream models respectively. And then we consider L2-normalised softmax scores as features to fuse them to make the final choice.

Figure 1: Two-stream logic embedding model

## 2 Experiments

### 2.1 Performance on single-relation neutral split of PGM dataset

Questions involving a single-relation type are easier than those involving multiple types to be solved. But experiments demonstrate that existing models behave poorly on the single-relation dataset. 3 shows that our model achieve a best performance on two single-relation types of PGM dataset at 95.5% and 98.6%, exceeding the best model existing when the datasets were published by 23.3% and 3.0%. These results, in particular, ones obtained in the shape type, are a testament to the ability of our model to do abstract reasoning. In fact, it is in the category of shape that these models struggle most, because the shape type of questions contains more attributions and rules.

### 2.2 Case Study

We do some more complicated experimenta on the PGM dataset. In this dataset, there are 29 types of questions in total, among which 20 types belong to shape type and 9 types belong to line type. We conducted three groups of experiments on shape type question and line type question respectively.

Table 3: Test performance of all models trained on the single-relation neutral split of PGM dataset. Teacher Model refers to adopt teacher model to provide the appropriate training trajectory. Type loss denotes the loss of question's category.

| Model | Shape(%) | Line(%) |
|---|---|---|
| LSTM[8] | 12.8 | 58.4 |
| CNN+MLP[8] | 15.3 | 60.3 |
| ResNet-50[8] | 25.6 | 61.0 |
| W-ResNet-50[8] | 28.9 | 62.2 |
| WReN [8] | 46.0 | 76.3 |
| LEN | 47.3 | 92.6 |
| LEN + Curriculum learning[1] | 40.3 | 90.2 |
| LEN + Self-paced learning[4] | 44.4 | 92.4 |
| LEN + Learning to teach[2] | 49.3 | 93.7 |
| LEN + Hard example mining[6] | 35.7 | 94.5 |
| LEN + Focal loss[5] | 41.1 | 86.6 |
| LEN + Mentornet-PD[3] | 58.3 | 92.9 |
| WReN + type loss[8] | 70.1 | 93.3 |
| LEN + type loss | 72.3 | 95.6 |
| LEN + Teacher Model | **79.4** | **97.9** |
| LEN + Teacher Model + type loss | **95.5** | **98.6** |

In the 2a, the left figure is the experimental results of shape type questions. "red" refer to that test performance on each type of questions after training on the whole single-relation PGM questions in random order. "yellow" refer to that test performance on each type of questions after training on each type of questions in random order. "red" refer to that test performance on each type of questions after training on the whole single-relation PGM questions with the guidance of teacher model. When training on the whole single-relation PGM questions together, various questions should promote each other to improve the accuracy. However, the distracting attributes make the test performance after training the whole single-relation PGM questions to be almost as accurate as the test performance after training on each type of questions. The accuracy in some types of questions is eveb 12.5%, which is consistent with the results of random guessing, which is caused by the existence of distracting attributes. However, the teacher model enable to perceive the appropriate trajectory for abstract reasoning task and achieve the best test performance.

(a)                                    (b)

Figure 2: LEN test performance on single-relation PGM questions, broken down according to the question's type. "red" refer to that test performance on each type of questions after training on the whole single-relation PGM questions. "yellow" refer to that test performance on each type of questions after training on each type of questions. "red" refer to that test performance on each type of questions after training on the whole single-relation PGM questions with the guidance of teacher model.