[Reviews · NeurIPS 2019]

Reviewer 1



The proposed architecture is novel but limited to the benchmarks. The curriculum learning has been already explored. Moreover, in terms of significance, how improving the performance in this task would be beneficial to real-world problems remains unclear. The presentation and clarity can be also improved please consider followings. - The description of the target task is unclear. - The definition of X_k at L151 is wrong. It confused me a lot and I thought that X_k is a set containing everything. I think the union operation should be replaced with a Cartesian product. - How can you be sure that the performance gain is from mitigating the effects of distracting features (L55)? The use of the curriculum does not directly mean mitigating the effects of distracting features. Please justify this argument as your paper is based on this argument.

Reviewer 2



Summary: This paper observes that the presence of distracting features is a key factor which causes substantial difficulties in abstract reasoning tasks. Inspired from the sequential nature of the human learning process, this paper proposes a feature robust abstract reasoning (FRAR) model which uses a reinforcement learning-based network as the teacher network and a Logic Embedding Network (LEN) that explicitly enumerates a much larger space of logic reasoning to disentangle abstract reasoning, as the student network. The authors further demonstrate the proposed model can have over 10% improvements over the state-of-the-art models on large-scale benchmark datasets, PGM [1] and RAVEN [2], in abstract reasoning. Originality: The main contribution of this paper lies in 1) its observation of one of the abstract reasoning’s fundamental characteristics: the co-presence of useful features for reasoning and distracting. Such observation provides an intuition why previous works on designing various kinds of models are not very effective for more sophisticated abstract reasoning tasks, and 2) following the key observation in 1), a proposed teacher-student framework that allows the teacher to learn to determine the appropriate teaching trajectory to optimize learner’s performance. A question I have is that how the task to distinguish reasoning features from distracting features differs from learning disentangled representations. Since many works have proposed their approaches to tackle the task of learning disentangled representations, it is recommended that the authors should also compare their proposed model with the approaches in previous works on learning disentangled representations (e.g. [3], [4], [5], etc.) under related work section. Quality: In general, this submission is technically sound, and the claims are mostly justified by the experiments. A few concerns are addressed below. The claims in this paper could be greatly strengthened by including a set of extrapolation experiments on the proposed model. Many real-world reasoning problems require the ability to extrapolate from limited training data (e.g., learning addition with 2-digit numbers and extrapolating to addition with 3-digit numbers). According to previous papers on abstract reasoning (e.g. [1]), the performance usually drops when the model is tested under extrapolation settings. As the performance on extrapolation is one of the key indicators of the model’s abstract reasoning ability and extrapolation can also be treated as one kind of distracting features, a set of experiment on extrapolation will further demonstrate the proposed model’s ability to distinguish distracting features and reasoning features. In addition, it would also be interesting to see how the performances of models other than LEN (e.g. RN, WReN, etc.) as the student networks can benefit from a teacher model. Clarity: The paper is generally well-written and structured clearly. Significance: This paper seems to be a useful contribution to the literature on abstract reasoning, showing a large improvement over the state of the art. =======================Post-rebuttal update============================== I am happy to main my ratings and recommend this work for acceptance after reading through other reviews as well as the author rebuttal, and engaging in the discussions. I look forward to seeing the discussions on disentangled representation, experimental results on extrapolation, and performances of models other than LEN in the camera ready version of this paper. [1] Adam Santoro, Felix Hill, David Barrett, Ari Morcos, and Timothy Lillicrap. Measuring abstract reasoning in neural networks. In International Conference on Machine Learning, pages 4477–4486, 2018. [2] Chi Zhang, Feng Gao, Baoxiong Jia, Yixin Zhu, and Song-Chun Zhu. Raven: A dataset for relational and analogical visual reasoning. In Proceedings of the IEEE Conference on Computer Vision and Pattern Recognition, 2019 [3] Hoogeboom, Emiel. "Few-Shot Classification by Learning Disentangled Representations." Msc, University of Amsterdam (2017). [4] Bengio, Yoshua, et al. "A meta-transfer objective for learning to disentangle causal mechanisms." arXiv preprint arXiv:1901.10912 (2019). [5] Hsu, Kyle, Sergey Levine, and Chelsea Finn. "Unsupervised learning via meta-learning." arXiv preprint arXiv:1810.02334 (2018).

Reviewer 3



Originality: The model proposed is quite novel. The teacher network uses recent development in reinforcement learning and applies in an innovative way to provide a curriculum for abstract reasoning task. While other curriculum learning works exist, this is one of the first to use it for challenging visual reasoning tasks. The related works regarding both abstract reasoning and curriculum learning are properly discussed and the authors do a good job of putting the proposed work in context of existing work. Quality: The proposed method is well-motivated and technically sound. The experimental setup is also clear and shows that both of the contributions (the LEN model and teacher model) are independently effective for both RAVEN and PGM dataset surpassing existing art by a solid margins (over 10% for both RAVEN and PGM). I also appreciate the authors including several other competing curriculum learning algorithms as well as testing the proposed teacher model on a variety of algorithms. Overall, this a a good quality submission. Clarity: The details are mostly clear. There are some minor writing issues. First, the use of past tense in abstract and body of text is unusual. E.g. "Table 1 illustrated such an idea..." instead of "Table 1 illustrates such idea". While it does not affect clarity, it reads odd. Most of the figures are blurry, I would consider using a vector graphics for revised version for better readability. Significance: As outlined above, the paper has two main algorithmic contributions. The teacher model shows success for a wide variety of models and is likely to be adopted for other tasks as well. Similarly, a large improvement in challenging visual reasoning task is also significant. The authors introduce a number of small task-specific engineering changes, such as the use of separate streams for processing "shape" and "line" features which are also likely to be useful to the community for similar visual reasoning tasks. *** POST REBUTTAL COMMENTS *** After carefully reading all other reviews, the author rebuttal and engaging in detailed discussions with other reviewers, I am happy to recommend this work for acceptance. The main merits of the work is in solving abstract reasoning datasets that have been independently studied (and duly peer-reviewed) to be a good proxy tests for the visual reasoning task. There is definitely valuable discussion to be had for how true that statement exactly is (as noted by R1). However, the authors of this submission should not have to defend the validity of community-established benchmarks. The author rebuttal also shows success in other tasks such as CLEVR. My ratings is unchanged from before: An accept.

[Author Response · NeurIPS 2019]

To reviewer #1:

**Broader applicability of the method.** Our method can be applied to a broader range of tasks that fall into visual reasoning. To name a few, Newtonian physics problem solving task [1], geometric problem solving task [6] and CLEVR as reviewer #3 suggested. We have observed a growing number of reasoning tasks in NLP such as multi-hop text question answering, knowledge graph reasoning and conversational models. Our approach can potentially be applied to strengthen the reasoning abilities of such tasks.

**Availability of the problem categories.** Category is widely available for abstract reasoning / visual reasoning which state-of-the-art models leverages as type loss. When such categories are missing, one can either get it using an unsupervised way such as the paper[5] that reviewer #2 suggested. Another way to deal with such a problem is to assume a latent category variable and optimizes it together with teacher model which could be a promising future work.

**Differences to [7].** We compared methods used in [7] and as shown in table 1 and table 2, it is not performing well for abstract reasoning. In [7], only the loss of the last step is used as opposed to the complete trajectory in our method. Additionally, in [7] action is taken to be a 0/1 decision on sample id but our action is a proportion of the related problems. All of these leads to the better performance of our teacher model. We will improve the paper to clearly outline the differences.

**Confirmation of the source of performance gain.** We have illustrated in the empirical study that training with a specific trajectory with different proportion of the distracting/reasoning features can dramatically improve model performance. This is illustrated in Figure 1, which shows that the difficulty of abstract reasoning task lies in the existence of distracting features. Table 1 shows that the appropriate training trajectory can greatly improve the model in the presence of distracting features. At the same time, the visualization of Sec 5.4 also shows that our method can distinguish distracting features better. We will put a visual training trajectory map in the final paper for better illustration.

**Other issues.** The reviewer is right that $\chi_k$ (L151) is a set of embedding of all triple-panels. We will fix it.

To reviewer #2:

**Disentangled representations.** Disentangled representation separates information on a single input while our method select inputs for a model. Our method is orthogonal to disentangled representation and can be applied on top of it. It is also worth mentioning that [4] actually implemented the idea of disentangled representation but with little improvements on abstract reasoning. One intuitive explanation is that distracting features live in a much more illusive manifold and disentangled representation along is not capable of separating it from reasoning features.

**Related work.** We will add discussions of disentangled representation into our related work.

**Extrapolation experiments.** We actually did an extrapolation experiment on the PGM. We separated training and testing in a way that they have non-overlapping values of "color" attribute and have achieved 8% improvement in accuracy. We are happy to include this result along with additional experiments in the final paper.

**Performances of models other than LEN.** We actually have a complete set of comparisons of WReN with/without teacher model in table 2. Performance of WReN is improved from 75.6% to 77.8% with type loss and from 62.8% to 68.9% without type loss. We will complete comparisons of other baselines (e.g., RN) in the final paper.

To reviewer #3:

**Performance improvements of the teacher network on LEN compares to WReN model on PGM.** The performance improvements on WReN is as significant as the one on LEN. Since the codebase of WReN has never been released, we implemented it on our own with an accuracy of 70.1% using type loss but we have never been able to reproduce the reported benchmark (i.e., 75.6%). Nevertheless, we include the performance of the published results of WReN in our paper for fair comparison. As a matter of fact, if we compare the improvements of teacher model against our own baseline on WReN the improvement is 7.2%. Comparing to 11.1% of improvements with LEN model.

**Testing on additional visual reasoning tasks.** We actually did an experiment on the CLEVR dataset but we didn't include it into the paper. our LEN model achieves 1.7% accuracy increase(95.5% to 97.2%) compared to RN[2]. Please note that CLEVR dataset is much easier than the datasets we used in the paper and the already high performance on baseline method allow only a marginal improvements using our teacher model. Nevertheless, the performance gain seems to be significant an consistent. We will explore the efficacy of the model on more widely used visual reasoning tasks (E.g., CLEVR-CoGeNT task) in the final paper.

**Writing Issues.** We will fix these issues as the reviewer suggested.

[1] Sachan M, et al. Parsing to programs: A framework for situated qa. KDD 2018.

[2] Santoro A, et al. A simple neural network module for relational reasoning. NIPS 2017.

[4] Steenbrugge, X., et al. Improving generalization for abstract reasoning tasks using disentangled feature representations. In Workshop on NIPS, 2018.

[5] Hsu, Kyle, et al. Unsupervised learning via meta-learning. arXiv 2018.

[6] Seo M, et al. Solving geometry problems: Combining text and diagram interpretation. EMNLP 2015.

[7] Yang Fan, et al. Learning to teach. ICLR 2018.


[Meta-Review · NeurIPS 2019]

Initially, this paper received mixed reviews. If I remember correctly, one reviewer was pretty negative to this paper especially mainly because of artificial task formulation. However, there was a long discussion between the reviewers after the rebuttal, all reviewers finally agreed that this paper has some value. So, I am happy to recommend accepting this paper as a poster.